# Changes of Material Elastic Properties during Healing of Ruptured Achilles Tendons Measured with Shear Wave Elastography: A Pilot Study

**DOI:** 10.3390/ijms21103427

**Published:** 2020-05-12

**Authors:** Borys Frankewycz, Leopold Henssler, Johannes Weber, Natascha Platz Batista da Silva, Matthias Koch, Ernst Michael Jung, Denitsa Docheva, Volker Alt, Christian G. Pfeifer

**Affiliations:** 1Department of Trauma Surgery, University Medical Center Regensburg, Franz-Josef-Strauss-Allee 11, 93053 Regensburg, Germany; leopold.henssler@ukr.de (L.H.); Johannes1.Weber@ukr.de (J.W.); Matthias.Koch@ukr.de (M.K.); Denitsa.Docheva@ukr.de (D.D.); Volker.Alt@ukr.de (V.A.); christian.pfeifer@ukr.de (C.G.P.); 2Experimental Trauma Surgery, Department of Trauma Surgery, University Medical Center Regensburg, Am Biopark 9, 93053 Regensburg, Germany; 3Department of Radiology, University Medical Center Regensburg, Franz-Josef-Strauss-Allee 11, 93053 Regensburg, Germany; Natascha.Platz-Batista-da-Silva@ukr.de (N.P.B.d.S.); Ernst-Michael.Jung@ukr.de (E.M.J.)

**Keywords:** Achilles tendon healing, Achilles tendon biomechanical properties, shear wave elastography

## Abstract

Therapy options for ruptured Achilles tendons need to take into account the right balance of timing, amount and intensity of loading to ensure a sufficient biomechanical resilience of the healing tendon on the one hand, and to enable an adequate tensile stimulus on the other hand. However, biomechanical data of human Achilles tendons after rupture during the separate healing stages are unknown. Shear wave elastography is an ultrasound technique that measures material elastic properties non-invasively, and was proven to have a very good correlation to biomechanical studies. Taking advantage of this technology, 12 patients who suffered from an acute Achilles tendon rupture were acquired and monitored through the course of one year after rupture. Nine of these patients were treated non-operatively and were included for the analysis of biomechanical behaviour. A significant increase of material elastic properties was observed within the first six weeks after trauma (up to 80% of baseline value), where it reached a plateau phase. A second significant increase occurred three to six months after injury. This pilot study suggests a time correlation of biomechanical properties with the biological healing phases of tendon tissue. In the reparative phase, a substantial amount of biomechanical resilience is restored already, but the final stage of biomechanical stability is reached in the maturation phase. These findings can potentially be implemented into treatment and aftercare protocols.

## 1. Introduction

Achilles tendon rupture (ATR) is a clinically relevant injury with an increasing incidence, which is associated with patients’ immobility and a significant socio-economic impact [1,2,3]. Therapy options vary from open surgery, to minimally-invasive surgery, to non-operative treatment, depending on type and localization of the rupture, individual patient-related factors, and the capabilities of the corresponding orthopaedic trauma facilities [4]. Without regard for operative or non-operative treatment, the rehabilitation therapy includes extensive physical, physiotherapeutic, and training protocols. In particular, a limited range of motion (ROM) is crucial in order to maintain the stump-to-stump distance to allow cellular and extracellular healing processes to repair and consolidate the damaged tissue. Data from biomechanical research show that an effective tensile load of the tendon leads to improved tendon healing [5,6]. To ensure a sequential increase of the tensile load, most therapy plans imply a successive increase of weight-bearing and cleared ROM within the first 6–12 weeks after rupture or surgery [7,8,9]. Physiologically, an appropriate balance between too little and too much load of the injured tendon is crucial to allow the tendon to heal without pulling the tendon stumps apart. Although these measures are routinely advised and implemented into aftercare, there is no consensus regarding the specific time phases for the subsequent clearance of ROM and weight bearing [10].

From the biological point of view, tendon healing occurs in three overlapping consecutive phases (inflammatory, reparative and remodelling (comprising of two stages: consolidation and maturation)) [11,12]. Biomechanically, an increasing gain of stiffness during the evolving healing phases is expected, but exact biomechanical values during the early stage of tendon healing are unknown. Although biomechanical data is available for the late stages of human Achilles tendon healing [13,14,15], data of early phases of tendon healing are limited to animal models [16].

Here, several contradicting time patterns of elastic behaviour during tendon healing were shown. In a murine Achilles tendon injury model, Steiner has shown that the tendons successively increased their strength during the reparative healing phase and reached 70% of their original stiffness after four weeks [17]. In a rabbit model Nagasawa et al. investigated the Young’s modulus throughout 12 weeks during tendon healing [18]. Similarly, the Young’s modulus gradually increased and reached nearly 80% of its uninjured value after 12 weeks. In a different approach, also in rabbits, Hirsch demonstrated a significantly improved stiffness just after 24 weeks of healing [19]. However, most animal injury models use a sharp transection as a method to sever the tendon, whereas in an actual Achilles tendon rupture the injury morphology is frayed and irregular. Therefore, their findings do not directly correlate to human healing scenarios, and their translational value is limited.

In an ideal experimental setting, precise biomechanically obtained parameters are desired to help developing physiotherapy regimes. For example, results from actual human in vivo biomechanical testing during bone healing in the 1970s gave insight into bone healing duration [20,21], which helped in creating contemporary weight bearing protocols [22,23]. However, such an approach is not possible in healing human tendons, because it would require explanting patients’ tendons. For the calculation of biomechanical parameters with the aid of dynamometers tendons need to be stable enough to undergo biomechanical testing. With this method Bressel et al. investigated patients, who suffered from ATR one to five years after injury [13].

Obtaining biomechanical data of early human Achilles tendon healing will give more insight about the durability and biomechanical competence of the healing tendon and will potentially enable the adaptation of more precise and more individualized physiotherapeutic rehabilitation protocols. Shear wave elastography (SWE) is a non-invasive ultrasound-based method for measurement of tissue material elastic properties. While the principles of shear wave excitation and detection are well established in structural and civil engineering [24,25], its application in the medical field was first implemented in magnetic resonance imaging (MRI) [26,27] before it was integrated in ultrasound technologies [28]. After establishing in other clinical fields, especially in liver diagnostics [29,30,31], SWE has begun to gain acceptance for musculoskeletal diagnostics. It has shown very good correlation to biomechanically tested tendons in animal models [32,33,34,35], human shoulder capsules [36], as well as in human Achilles tendon cadavers [37]. Several groups have used SWE experimentally to characterize tendon pathologies in the late stages of tendon healing [14,38,39] as well as in tendinopathy [40]. Under the assumption of a linear isotropic model, propagating shear waves, created by the ultrasound transducer, are measured inside the tendon. Their propagation speed (shear wave velocity = SWV) corresponds to the Young’s modulus of the tissue [41], which describes the material elastic properties. Due to its non-invasive character it can be used during of the whole process of tendon healing.

In this pilot study the biomechanical properties of healing Achilles tendons after rupture were prospectively analysed using SWE. It was hypothesized that the material elastic properties of the ruptured area would increase with the progressing stages of tendon healing throughout one year after injury.

## 2. Results

### 2.1. Patients

Twenty-seven patients met the primary inclusion criteria and 15 of those agreed to participate in the study. Three of those were excluded due to lost-to-follow-up, so 12 (11 male and one female) patients could be included in the study. The mean age of the patients at the time of injury was 39.9 ± 12.2; (21–62) years. All of the injuries occurred via an indirect trauma, of which 11 occurred during increased physical activity and one due to a gait irregularity. All patients had a complete tendon rupture. Mean distance of the rupture from calcaneus was 4.6 ± 0.84 cm. Three of the 12 patients were treated operatively (O) and nine non-operatively (NO). None of the patients had a re-rupture. No rehabilitation- or orthosis-related problems or complications were reported to occur. Appendix A gives an overview of the included patients, their demographic data, relevant comorbidities, SWE results and elastographic curves.

### 2.2. Shear Wave Elastography

Figure 1 shows SWE images with a typical course of increasing material elastic properties. Shear wave velocity (SWV) is noted in m/s. Internal consistency of the sonographic examination was very good with a Cronbach’s ∝ of 0.96 for insertion, 0.98 for distal, 0.98 for rupture, and 0.96 for proximal area, respectively.

### 2.3. Elastographic Behavior of Healing Tendons

Due to the low number of operatively treated patients (*n* = 3) the evaluation of this group was limited to descriptive analysis (data shown in Appendix A). In the main study group (NO) all of the scanned tendons showed, at time of injury, lower SWV values in the rupture areas compared to their healthy contralateral tendons (*p* = 0.0156). All scanned areas except the insertion showed an immediate SWV drop compared to baseline. The more proximally the tendon was measured, the lower the drop. After one week, SWV increased gradually within the first six to nine weeks. At the rupture site as well as in the adjacent proximal and distal area, there was a significant increase between week 1 and 3. That increase continued until week six (significant in the rupture area), where it reached a plateau. Up to this time point, the differences were still significant compared to baseline values: 80% of elastographic stiffness was achieved at week 6 and 90% at week 12 (compared to baseline). At the insertion site the SWV decrease occurred with a one-week delay. It was not as substantial as in the mid-tendon areas but also reached a plateau phase after six weeks. A second increase of SWV occurred three to six months after injury, which was significant in the ruptured area. In the contralateral (healthy) tendons, a delayed decrease was observed, which was significant to baseline at week 12 at the rupture site and at months 6 and 9 at the insertion. Figure 2 and Figure 3 show the elastographical behaviour of all groups at the rupture site and insertion (Figure 2) and the distal proximal and distal areas (Figure 3). The elastographic behaviour of one patient with three relevant comorbidities (Patient #7: diabetes, polyneuropathy and pAOD) suggests a lowered plateau between the sixth and twelfth week (see Appendix A), but no statistical correlation was found between the presence of a comorbidity and SWV at these time points.

## 3. Discussion

In this study, the material elastic properties of ruptured human Achilles tendons were determined with shear wave elastography. An increase of material elastic properties in two phases was found during a healing period of one year. Prior to this study, insight into biomechanical parameters during the early phases of tendon healing were limited to animal models, which show different patterns of elastic behaviour [17,18,19]. Investigating human healed Achilles tendons after rupture one to five years after injury, Bressel et al. found differences in isometric torque and peak passive torque, but no differences in stiffness [13]. However, measurements with this method can be biased as they refer to the muscle-tendon-complex and not the tendon only [13,42].

SWE is an ultrasound method that allows measuring material elastic properties of a designated area within a certain tissue depth of echogenic tissues. Shear waves are being created in the tissue of interest and their propagation velocity (SWV) can be measured. Shear waves propagate with ~1–50 m/s and their velocity is a direct indicator of material elastic properties [41]. With the knowledge of the density (ρ) and the Poisson’s ratio (ν) of the measured body, V can be transformed to the shear modulus (G, with the unit kPa) and Young’s modulus (E, also in kPa): E = 2G (1 + ν) = 2 ρV2 (1 + ν). Some SWE machines express the measured values in kPa, assuming constant ρ and ν values. Both m/s and kPa units are found throughout the literature. They have different values, but both express the relation of stress to strain, which describes the linear elastic properties of a body [32,41]. The physical term “stiffness” technically requires knowledge of the geometry of a body and describes the measure of the resistance offered by an elastic body to deformation [43]. In the context of SWE, geometric properties are unknown and the measurement of material properties (relation of stress/strain independently of geometry) are desired. Therefore, the measured SWV, which technically expresses the material elastic properties of the assessed object in m/s, will be referred to as “elastographic stiffness”.

In order to validate the correlation of SWE to actual biomechanical values, a number of comparative studies have been carried out [32,33,34,35,36,37]. In regard of the human Achilles tendon, Haen et al. used 11 human Achilles tendon cavaders to correlate SWE with simultaneously biomechanically tested tendons. With a correlation coefficient of R^2^ = 0.95 ± 0.05 (*p* < 0.005) SWE has proven a very good correlation. Regarding the ultrasound machine that was used in our study, Rosskopf et al. found a strong positive correlation of SWE measurements and values from biomechanical testing of bovine flexor hallucis longus tendons (Pearson’s r = 0.877–0.915, *p* < 0.001) [32].

SWE has been used in a few studies to determine material elastic properties in healing human Achilles tendons. Chen et al. investigated freshly ruptured Achilles tendons and found similar low SWE values [44]. Using a different machine manufacturer (AixPlorer, Supersonic Imagine, Aix-en-Provence, France) their study showed that the tendon stumps and the freshly hematoma within 24 h after rupture revealed values approaching zero kPa. Tendons in the advanced healing stages (*n* = 2) showed also comparatively increased values (±44 kPa). Zhang et al. have examined ruptured Achilles tendons in the late stages of tendon healing in 26 operatively treated patients [39]. At 12, 24 and 48 weeks after rupture a gradual increase of the Young’s modulus was found. These data bear a resemblance to our findings, although the relative percentage of the baseline values were to some degree delayed: 187.7 ± 23.8 kPa at 12 weeks (approx. 63% of baseline), 238.3 ± 25.3 kPa (approx. 79% of baseline) and 289.6 ± 23.4 kPa (approx. 96% of baseline). This can be due to different baseline approximations and also to the more restricted physiotherapy protocol after surgery.

In this pilot study the material elastic properties of ruptured human Achilles tendons were investigated during the course of the first year of healing. Due to the small number of operatively treated patients the focus of the analysis was put on the non-operatively treated patients. The SWE values scaled on the one-year time-plot express a bi-sigmoidal pattern. The first and largest increase of elastographic stiffness was found within the first six weeks. The second and less substantial one was found between nine weeks and six months after injury.

When observing the elastographic behaviour of the tendon with regard to the biological tendon healing phases, an interconnected pattern can be described as follows: as expected, the elastographic stiffness drops instantly in the rupture area after the injury (inflammatory phase). In this phase, the gap between the stumps is filled with liquid hematoma from ruptured vessels and capillaries from the endo- and epitendineum as well from the adjacent peritendineum [11,12] (see Figure 1). The tendon fibres are frayed and discontinued. Here, within the first week after injury the lowest elastographic stiffness was observed, especially in the ruptured area and distally of it. In the following reparative phase (±6–8 weeks), fibroblasts start producing predominantly collagen III in high amounts, providing a first reticular network [11,12,45]. Interestingly, the highest increase of elastographic stiffness was found already during these first six weeks after injury. Biochemically, collagen III fibers connect multilaterally to each other over covalent bonds, and thereby stabilize the primary fibrous scar tissue [46]. In interaction with proteoglycans and other ECM components, the forming of a “tendon callus” is orchestrated, which appears to go along with an incremental consolidation of biomechanical properties. After six to eight weeks the remodelling phase starts (consolidation phase), and collagen III is successively replaced by collagen I. The ECM structure starts transforming, the cell/matrix-ratio decreases and the collagen fibres start changing their alignment along the direction of the tensile force [47]. In this phase, the elastographic stiffness seems to reach a plateau, resembling the ceiling effect of the reorientation processes. In the maturation phase, which according to Tillman et al. starts at about ten weeks after injury [48], the cell metabolism declines and collagen cross-linking increases, transforming the tendon callus into a more mature tendinous scar tissue. Findings from this study show that during this last phase, another increase in elastographic stiffness occurs. This can be an effect of the increased amount of collagen cross-links [49]. Even though the exact mechanisms and correlations of biochemical processes and changes in biomechanical properties during human tendon healing are not identified, this study suggests some first insight into a possible association of the four healing phases and biomechanical behaviour.

Interestingly, SWV at the calcaneal insertion decreased with a week delay. The decrease was not as substantial as those at the sites closer to the rupture, which can be explained by the persisted continuity of the tendon tissue at the insertion. However, the observed decrease resembles the lack of tension during the first three-week-period, where the upper ankle was immobilized in 30° plantar flexion and only partial weight-bearing was allowed. These observations are consistent with prior measurements of plantar-flexed ankles [50]. Additionally, the lack of tension leads to a matrix degradation. Gimbel et al. have shown in a murine rotator cuff model that the structural tendon tissue degradation start as early as one week after tendon detachment [51]. That degenerative process most likely goes along with the loss of capacity of resistance to stress, which is reflected in the decrease of the Young’s modulus slope [12]. A relation of shortage of load and decrease of elastographic stiffness was shown before [52,53]. This supports the hypothesis, that a tensile load is necessary to maintain the biomechanical properties [16].

In contrast, the proximal area of the Achilles tendon showed a delayed increase in elastographic stiffness in the sixth to ninth week post injury. Possibly, the load required to increase SWV during healing is more distributed on the wider diameter of the tendon in this area [54] and, therefore, the stimulus might be scattered.

The indication for a non-operative treatment pathway presupposes an adequate adaptation of the tendon stumps in 20° plantar flexion and a gap of maximally 1 cm in neutral ankle position [55,56]. After application of an immobilization cast in 30° plantarflexion, the required contact of the tendon stumps is assured throughout the first three weeks of tendon healing. After three weeks, an influence of the successively increased ROM on the elastographic stiffness can be questioned. Since the measurements were taken before the adjustment of the orthosis and in a relaxed position without additional tension, the immediate effect bias was obviated. However, with the first adjustment of the orthosis after three weeks the tendon had to adjust to the new force constellation (additional stress). Since SWE describes the relation of stress and strain, the equivalent elastographic stiffness resembles the resilience of the tendon tissue to stress forces [57]. An increase of elastographic stiffness after the next three weeks demonstrates that the additional stress probably even enhances the capacity of resisting to stress (stress/strain ratio increases). This observation might be an argument for early and progressive physiotherapy. Recent studies propagate a more radical allowance of weight bearing and range of motion. In a meta-analysis of post-operative rehabilitation protocols Braunstein et al. found that none of the progressive schemes resulted in a worse outcome or increased re-rupture rates [58]. Under the premise that the first few months after rupture are crucial for post-traumatic function, the authors suggest an immediate full weight-bearing and a restricted plantar flexion in 20° for only two weeks. Similarly, Schepull et al. found a higher elastic modulus at 19 and 52 weeks after surgery when comparing patient with continuous cast immobilization vs. patients with controlled tensile loading [59]. Although these studies investigated operatively treated patients, the data indicates that early tensile loading improves the mechanical properties of the healing Achilles tendon.

A major limitation of this study is that biomechanical values cannot be sufficiently correlated with micro-morphological and organic content parameters in a human setup. Obtaining biomechanical data, as well as healing human tendon tissue for morphological and content analyses, is ethically not possible, because it would require explanting patients’ tendons in different healing phases. However, data from this study suggest a relationship of biomechanical behaviour and the healing stages.

A conceptual limitation of the study was that the contralateral side was not scanned at weeks 6 and 9. That was because a decrease was not expected at those time points. This study revealed, though, that especially within the first 12 weeks after injury, the contralateral tendon decreases in elastographic stiffness. Most likely this is due to the general reduced mobility. An increase in elastographic stiffness was shown for short-term load (already after 30 min of running) [52] and also between physically active people (six or more hours of weight-bearing exercise per week) and a normal population [53]. One would expect that the contralateral limb would be more loaded due to the limited weight bearing on the injured limb but probably the activity level of injured patients declines to a general low level. These considerations and questions should be accounted for in succeeding full-scale studies.

Another limitation was the small sample size, which leads to a high variability and, consequently, to high standard deviations of the time points. Physiological healing processes underlie numerous individual intrinsic and extrinsic factors that can neither be eliminated nor sufficiently influenced in the clinical setting.

A technical limitation was the peak measurement threshold of the ultrasound machine. Especially in the surgically treated patients the maximally high SWV was measured after half a year at the distal part of tendon and the calcaneal insertion. These values correspond with physiological tendon values that were measured by Fu et al. with the same Siemens Acuson 3000 machine [60]. Shear-wave-elastography machines from other manufacturers (e.g., AixPlorer, Supersonic Imagine, Aix-en-Provence, France) offer a higher peak measurement threshold but the results (SWV, given in m/s) cannot be directly compared due to different shear wave excitation technologies [32]. However, none of our patients had a baseline measurement (contralateral healthy tendon) of more than 10.0 m/s, therefore the calculations of the baseline ratios in this study are realistic.

Concerning the biomechanical expressiveness of SWE, it has to be taken into account, that this ultrasound method is only a snapshot of the elastic modulus (stress/strain ratio) of the tendon, in this case of a relaxed tendon [12,61]. Since a strong correlation of elastographic measures and the clinical outcome was not shown (see Appendix B), the results of SWE may not be interpreted as a definite healing parameter but rather as an indicator for biomechanical stability. Slane et al. have shown that SWE values differ in differently preloaded tendons [62]. Since SWV and the Young’s modulus are equivalent, this is in accordance with the physiological stress/strain behaviour of a tendon, where the ratio increases from the “toe region” (relaxed tendon) into the linear elastic region (stressed tendon) [12,32,61,63]. In an early post-trauma situation, especially in non-operatively treated patients, additional load in the rupture area is undesired because of a re-rupture risk. However, for future examinations of mainly operatively treated patients without load and with a certain additional minimal preload might be considered to give better insight into the individual tendon healing behaviour.

In summary, this pilot study suggests a time correlation of biomechanical properties measured by SWE ultrasound with the biological healing phases of tendon tissue, showing a substantial restoration of biomechanical resilience in the reparative phase and an additional stabilisation in the maturation phase. Since SWE is an imaging technology purely based on detection of micromechanical structural properties, it needs to be interpreted cautiously. However, since biomechanical testing of human Achilles tendons is not practicable, SWE allows an effective and feasible estimation of biomechanical properties. For clinicians it is an important issue to connect biological and biomechanical changes during healing with clinical aspects, in order to strive for optimized outcome. However, the clinical value of SWE requires further investigation in large-scale longitudinal studies.

## 4. Materials and Methods

### 4.1. Patients

Patients who suffered from ATR and presented in the Emergency Department of the University Regensburg Medical Centre (Regensburg, Germany) between February 2016 and February 2018 were acquired for the study. Primary exclusion criteria were bi-lateral injury, prior ATR in patients’ history, arthrodesis of one of the upper ankle joints or other diseases causing immobilization, neuropathic or malignant diseases, age <18 years or a >48 h delay in emergency room examination after the rupture. Participation in our standardized therapy and rehabilitation protocol was obligatory as well. Secondary exclusion criterion was re-rupture within one year. Diagnosis was made based on B-mode ultrasound, pursuant to the Amlang criteria [64]. A lateral view X-ray was conducted to exclude an avulsion fracture. Decision about whether operative (O) or non-operative (NO) treatment was made by the supervising attending trauma surgeon, based on the generally accepted and recommended treatment guidelines [55,56]. In total, 12 patients could be acquired for this study. All participants provided written, informed consent prior to voluntary participation. For evaluation of the clinical outcome patients completed the Foot and Ankle Outcome Score (FAOS) [65] at time of each examination. The study was performed after approval of the University’s Ethical Committee (Ethical Grant Number 15-101-0019, approval date: 26 March 2015).

Surgery on patients that were selected for operative treatment (O) was performed with standard open surgery technique by one supervising attending trauma surgeon. Briefly, a peritendinous medial incision was made and the paratendineum was split longitudinally. Hematoma and debris was removed and the tendon stumps were retracted under plantar flexion. A standard Kirchmeyer–Kessler-suture was applied to maintain stump reduction using a PDS suture size 1. The peritendineum was sutured as well. Success of reduction was controlled visually by careful dorsal extension movement and a negative Thompson test.

### 4.2. Rehabilitation Program

All patients from both operative and non-operative groups received a standardized rehabilitation program. It included immobilization in an VACOPed® orthosis (OPED GmbH, Valley/Oberlaindern, Germany) in 30° plantar flexion for day and night and 20 kg weight-bearing for the first 3 weeks of rehabilitation. From week 4 to 6, the plantar flexion angle of the orthosis was reduced to 15° and weight-bearing was increased to half bodyweight. From week 7 to 8, the upper ankle joint was immobilized in a neutral position for only daytime and the patients were allowed full weight-bearing. At this time point, careful passive mobilization of the upper ankle joint was conducted by a physiotherapist. After 9 weeks, the orthosis was removed and isometric training of the calf muscles was implemented without active stretching of the triceps-surae-complex. Up to this point, the patients received lymph drainage on a regular basis. After 12 weeks, the patients were allowed to participate in straightforward sports (e.g., swimming, running, biking) and start with careful active stretching. At the same time, they were instructed to avoid stop-and-go and contact sports (e.g., soccer, squash, tennis) up to six months postoperatively.

### 4.3. Ultrasound Examination

For assessment of the material elastic properties of the Achilles tendons a commercially available Acuson S3000 ultrasound system (Siemens Healthcare GmbH, Erlangen, Germany) coupled with a linear array transducer (4–9 MHz) was used. All measurements were performed using the Virtual Touch Imaging and Quantification mode (VTIQ) by two medical staff members who underwent specialized training for this purpose. Achilles tendons were scanned in a prone position with a fully extended knee and the foot hanging over the edge of the examination bed in a force-neutral position to avoid tendon stress [66]. The injured tendon was examined at the first day of presentation in the ER and 1, 3, 6, 9, 12 weeks as well as 6, 9 and 12 months after injury or surgery, respectively. All ruptured tendons were scanned in the sagittal plane view at four areas: the rupture site (“rupture”), the area very proximal of the rupture (“proximal”), the area very distal of the rupture (“distal”), and at the calcaneal enthesis (“insertion”). The Region of Interest (ROI) was selected by using the B-Mode image for primary orientation. Within each ROI, 5 to 15 measuring points were randomly selected to generate quantitative data (shear wave velocity (m/s); see Figure 1). The SWE scans were repeated at least twice in order to obtain a mean value of the shear wave velocity. Values before the rupture were not available, therefore the contralateral non-injured tendons (CL) were scanned at the first examination and served as estimated baseline values. Additionally, the contralateral tendon was measured after 3 weeks and after 3, 6, 9 and 12 months, 6 cm proximal of the calcaneal insertion (“tendon”) and at the insertion itself (“insertion”).

### 4.4. Data and Statistical Analyses

Results are given as mean ± SD; (range). Descriptive curve analysis (fitted curve) was performed to analyse the biomechanical behaviour during tendon healing. To test for significant changes during tendon healing, differences between each consecutive scan time point were detected using the Wilcoxon signed rank test in every scanned area, respectively. To compare the injured tendons to baseline values (CL) the Wilcoxon signed rank test was used for the rupture area and insertion at all time points. Values from the proximal, rupture and distal site were averaged and normalized to baseline to estimate the percentage of original material elastic properties. Reliability (internal consistency) of the measurements (averaged value per image) was estimated using Cronbach’s ∝ for each scanning area. Values of *p* < 0.05 were considered statistically significant.

## 5. Conclusions

In this study the material elastic properties of healing Achilles tendons were investigated prospectively during the first year after rupture by SWE ultrasound. A bi-sigmoidal increase of elastographic stiffness was found, suggesting a possible concordance with the biological healing phases. A significant increase was found within the first six weeks of tendon healing, indicating an early restoration of the biomechanical properties in the early stages of tendon healing (reparative phase). Even though SWE is not an exact measurement of stiffness, but an estimative technique of assessing material elastic properties, it can serve as a straightforward and non-invasive tool to monitor functional healing processes. It has the potential to help in decision-making of optimized and individualized physiotherapy protocols. However, more comprehensive large-scale studies are necessary to implement detailed normality values, healing response clusters and potential individual factors like level of physical activity and comorbidities.

## Figures and Tables

**Figure 1 ijms-21-03427-f001:**
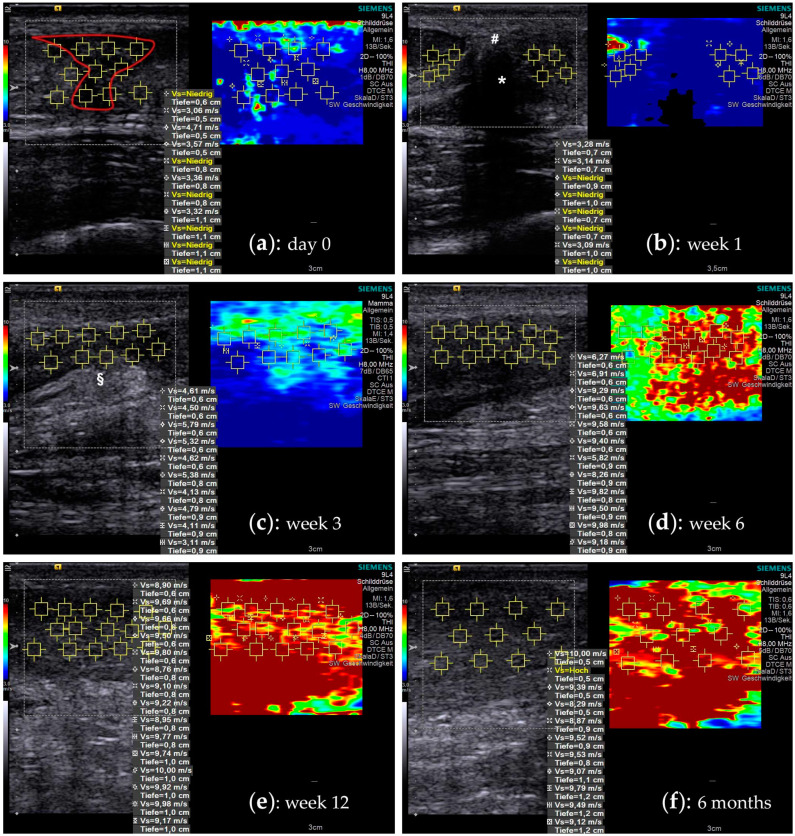
Shear wave elastography images during the course of tendon healing. A split screen view with a B-mode ultrasound image on the left side allows choosing a particular ROI (yellow squares), where the SWV is measured. A corresponding color-coded SWE image on the right side visualizes the elastographic stiffness. (**a**) The figures show a time-course of the rupture area of a 26-year-old male patient that was treated non-operatively: at the time of injury the tendon gap is filled with hematoma (red outline). The tendon stumps are retracted and due to the lack of tension, they show a very low SWV, which often is below the measurement threshold (“Vs = Niedrig”). (**b**) At one week the hematoma is mostly consolidated (#), connecting the remnants of the original tendon substance (* acoustic shadowing). (**c**) At three weeks, the hematoma is resorbed and the primary tendon callus, with partly hyperechoic fibrous tissue (§), starts regaining its biomechanical strength. (**d**) Six weeks into the healing process, the tendon callus has reached a significant increase in elastographic stiffness. (**e**) In the following remodelling phase, the tendon tissue further consolidates, expressing an additional increase (week 12). (**f**) In later phases the measurements often reaches the peak measurement limits of 10 m/s of the ultrasound machine.

**Figure 2 ijms-21-03427-f002:**
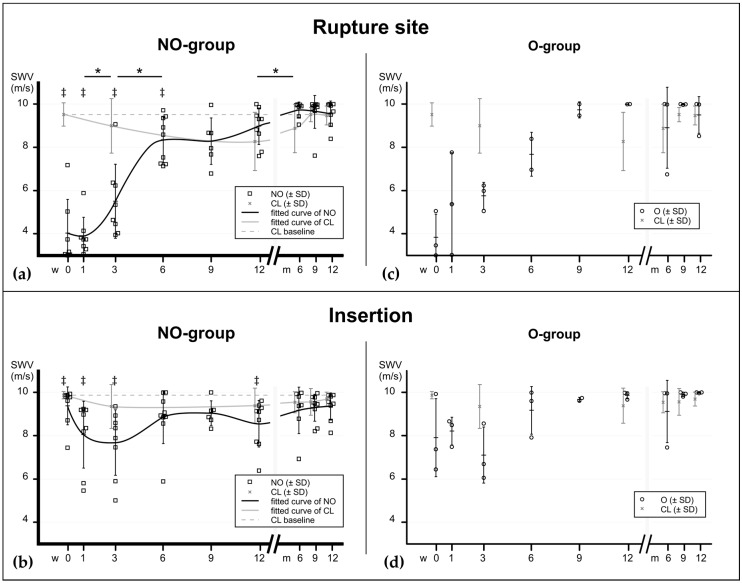
Material elastic properties during tendon healing at the rupture site and insertion, assessed with SWE, expressed in m/s. The graphs show the progression of elastographic stiffness in relation to time after injury (0–12 weeks and 6–12 months). (**a**) In the mainly invested group—the non-operatively treated patients—the elastographical pattern at the rupture site shows a bi-sigmoidal increase (fitted curve line) with a major increase between the third and the sixth week, suggesting an early regaining of material elastic properties. Another less substantial increase occurs after the ninth week, at the end of the consolidation phase. Interestingly, the biomechanical properties of the contralateral tendons (grey line) drops during the first 12 weeks before increasing back to baseline values. (**b**) At the insertion site the decrease of material elastic properties is less substantial due to the continuity of tendon tissue in this area. Significant differences between the consecutive time points of the NO-group are noted with asterisks (*). ‡ show significant differences between the ruptured tendons (NO) and baseline values (CL at time of injury, dashed baseline) at the corresponding time points. *p* values: * ≤0.05. (**c**,**d**) Elastographic values of operatively treated patients for comparison. NO = non-operatively treated patients, O = operatively treated patients, CL = contralateral tendons (pooled).

**Figure 3 ijms-21-03427-f003:**
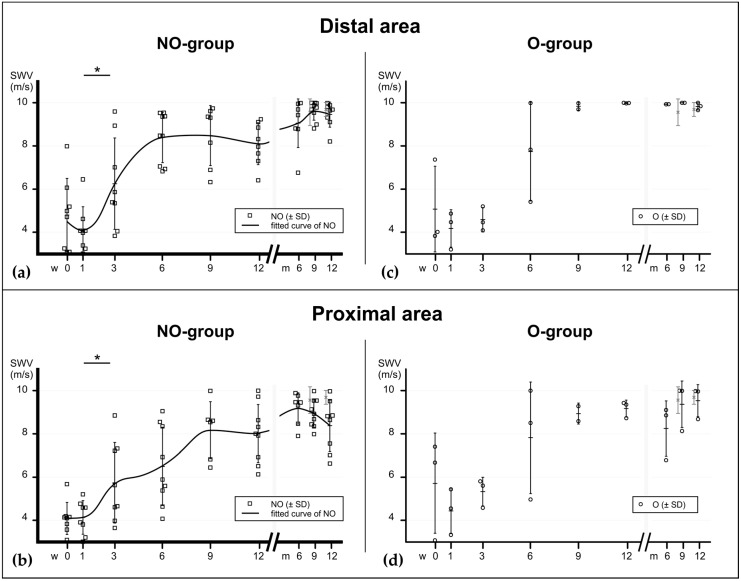
Material elastic properties during tendon healing at the area very distal and very proximal to the rupture, assessed with SWE, expressed in m/s. (**a**) In the mainly invested group—the non-operatively treated patients—the elastographical pattern at the distal area also shows a bi-sigmoidal increase (fitted curve), similarly to the rupture site. (**b**) In the proximal area the increase appears to occur more delayed. Significant differences between the consecutive time points of the NO-group are noted with asterisks (*). *p* values: * ≤0.05. (**c**,**d**) Elastographic values of operatively treated patients. NO = non-operatively treated patients, O = operatively treated patients, CL = contralateral tendons (pooled).

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
