# Peer review of "Changes of Material Elastic Properties during Healing of Ruptured Achilles Tendons Measured with Shear Wave Elastography: A Pilot Study"

_ijms, 2020, doi:10.3390/ijms21103427_

Round 1

Reviewer 1 Report

The manuscript "Changes of elastic properties during healing of 2 ruptured Achilles tendons measured with shear wave 3 elastography: a pilot study" has been modified according to previous suggestions. The topic of the study is interesting and focus on the use of shear wave elastography (SWE) as a non-invasive ultrasound-based method to measure the tendon elastic properties. However, it is still completely preliminary and needs of further evaluations to better clarify its efficacy and reliability as a measure of the biomechanical properties. I recommend authors to consider it for further future studies.

Overall, in my opinion this manuscript needs some minor improvements before being accepted. I still have some concerns about the figures (2 and 3). I do not think the results are easily understandable as they are. Authors should improve both graphs, by separating the information reported (operatively treated and non-operatively treated groups should be separated) and explaining it better both in the figure legend and in the results paragraph. I also believe there is some statistical inaccuracy. In M&M the authors reported that the results are expressed as the mean ± SD. However, in the figures it is not so.

Reviewer 2 Report

This paper describes the study carried out on several patients with their Achilles tendon ruptured. The shear wave elastography technique was used for such study, by correlating its results with the healing phases.

The shear wave elastography is a novel and promising technique under investigation only few years ago. Some results have appeared recently on researching papers and could be a widely used technique in the future.

  In the Patients section (paragraph 4.1) isn’t said the number of studied patients, and this number should appear. Only in the abstract is said that were studied 12 patients after an acute Achilles tendon rupture, three of them suffering a surgery. This number of cases looks quite short to obtain significant results, and in any case, it should appear in the methods section.

  Besides the low number of studied cases, the weakest point of this paper, according my opinion, is that the study is only done qualitatively, with only a few quantitative results.

  In the Results section (paragraph 2.3) are given the results of the study as curves relating the speed of the waves (in m/s) against the weeks/months after the injury (figures 2 and 3), and also are given the images of the study (figure 1). The caption of figures describes the graphs as “Elastographic stiffness during….” But the quantity graphed is a speed (m/s) and not a stiffness (in KPa or MPa or even in N/m). It’s true that on the Discussion section (lines 186 and 187) are given a few data in kPa, but these data appear without any explanation. Have been these data given by the measurement device? Have they been calculated? How? The translation from data of speed to stiffness should be explained and the data and results should also clearly be given.

  The evolution of the curves are commented and related to the healing phases of the tendon, but this explanation would require numerical values. In my opinion is not valid to suggest relationships between the behaviour of the experimental curves and the healing stages without measurable results. That could be an interesting result, as it is discussed on lines 195 to 219, but not only in a qualitative way. As a suggestion, this study could be improved by studying, i.e., different rehabilitation techniques, and measuring the stiffness of the tendon along the healing process with the different techniques. Such study could asses some of the hypothesis given in the paper.

 On the other hand, on lines 174 to 176 is said that the term “elastographic stiffness” is used instead the Young’s modulus because it requires the knowledge of the geometry. Therefore, the units of the “elastographic stiffness” shouldn’t be KPa or MPa (units of Young’s modulus). This discordance should be clarified.

  Besides the before comment, the organization of the paper is not the best: the first part of the Discussion (paragraph 3, lines 146 to 184) should probably be placed in the introduction.

 There are also two similar sentences (lines 226 to 227 and 285 to 287) about the stress strain curve of the tendon, but I can’t understand the meaning of them. Please, clarify these sentences.

I feel that this paper shouldn’t be accepted in this way. Only after a major revision, by incorporating numerical values, rewriting discussion and results sections, could this paper be accepted. Perhaps a short communication or something like that would be a more appropriate type of paper.

Reviewer 3 Report

The authors evaluated non-invasively changes of elastic properties during healing of ruptured Achilles tendons measured with shear wave elastography. They found a bi-sigmoidal increase of elastographic stiffness. The method offers the opportunity to describe biomechanic properties during the healing process. The research topic matches well with the theme of the special issue of the journal. Despite only few patients could be included the data are convincing and well presented. The supplemental data give profopund additional information. The manuscript is well written.

Line 55: are the healing phases rather overlapping?

Line 71: Achilles tendon (AT) is abbreviated here but the abbreviation is not used in the following text. Later Achilles tendon rupture is abbreviated „ATR“ but also not consistently used.

Line 90: „39.9±12.2;[21-62]“ insert a blank

Line 93: show some key data as a regular table within the main body of the manuscript

Line 156: remove the surplus „is“

Line 173: „form“ means „from“?

Line 202: add „fibroblasts“ since they are more synthetically active

Line 199: how about the epitendineum/ epitenon?

4.1. patients

Provide some more information in regard to the exact ATR localization (see suppl data, some key Information could be transferred), was it always a complete rupture?

Please provide, if possible information about the presence or Absence of comorbidities which could influence healing (e.g. diabetes mellitus, hyperparathyroidsm, kidney disease)

Reviewer 4 Report

In their study, the authors perform ultrasound elastography in 11 patients after achilles tendon rupture, in order to determine changes in stiffness during tendon healing. Despite the relatively low number of patients included, the study is well conducted, the manuscript is well written.

Minor comments:

line 154: "healing human tendons"

Supplementary material: The table is slightly confusing, patient data should be graphically more separated from the time points described

Round 2

Reviewer 2 Report

I see the authors have understood my review and they have accordingly modified the text. Even though I think this paper shoulb be accepted, there are still two comments I would like to do:

The first one is about the term “elastographic stiffness”. After reading your answer I see that we share the opinion in some way. Physically, this term is undefined. Probably, it would result correct to speak about “stiffness material properties” by referring to a set of physical parameters as Young’s modulus, shear modulus, … but “elastographic stiffness” is not a physical parameter, and that’s why it results difficult for me to accept the caption of figure 2. From such caption it results evident to correlate the physical parameter elastographic stiffness with a speed, and that’s not correct. May be, if you directly write the equation you show on your response relating Young’s modulus, density and shear-wave speed, it could clarify this point, because it highlights that the shear speed is not equal to Young’s modulus.

An the second comment is about the difficulty to relate numerical values and healing phases. I recognize the difficulty, but it is always desirable to obtain objective and quantifiable conclusions.

Nevertheless, I think this is valuable paper working with a new technique and that can be interesting for the clinician people.

Author Response

Dear Reviewer,

Thank you for re-evaluating our manuscript and responses, we highly appreciate it.

We appreciate your plea for correct terminology of physical parameters. Indeed, the nomenclature of SWE expressed in m/s is challenging. We have discussed and consulted this topic one more time with our mechanical engineer. Biomechanically speaking, the correct naming of the measured properties using SWE is “material elastic properties”. Based on your suggestions, we have changed the naming in a) the captions of figures 2 and 3, b) the title, c) the abstract, and d) throughout the manuscript, accordingly (changes to the newly submitted manuscript appear in blue font).

      Also, as suggested, we have added the equation to the discussion paragraph, in order to make the relation and conversion of SWE to Young´s modulus better understandable (196-198). Thank you for that suggestion.

      Since physically speaking “stiffness” describes geometric elastic properties and there is no physical term that describes material elastic properties in the context of SWE correctly, the introduced term “elastographic stiffness” helps to simplify the complex terminology. For a better description, we have added another explanation in the definition of this term, in order to clarify the context of SWE (measured in SWV and expressed in m/s) and material elastic properties (205-206).

This manuscript is a resubmission of an earlier submission. The following is a list of the peer review reports and author responses from that submission.

Round 1

Reviewer 1 Report

Authors present a radiologic ultrasound study where they measure shear wave velocity in Achilles tendons during healing. They conclude on tendon elasticity from these values.

This study design deserves several criticisms: There is no validation if SWV really reflects elasticity, at least the authors cannot cite any. Are there biomechanical studies that can prove this correlation? Only ultrasound values are reported, and they cannot be transformed into mechanical parameters one to one, so interpretation of this data must be done with caution. 

And from a clinical point of view, one would have to know breaking strength to plan a physiotherapy regime. SWV is just too vague to base any therapies on it.

This work is suitable as a pilot study to plan further investigations, but not as publication in a high-ranked molecule biology journal. Neither the journal nor the quality of the work are adequate.  

Reviewer 2 Report

The manuscript is well structured and the results presented provide significant insights into understanding the tendon healing process. Although preliminary, this study offers an innovative approach based on the use of ultrasound with the shear wave elastography technique to measure the elastic properties of the tissue, thus offering a clearer view of the specific healing phase that the tissue is experiencing.

I have no major comments to provide, except for a suggestion regarding the way the authors decided to graphically present the measurements obtained (Figure 2-3). I’d suggest them to use a different type of graph or to use colors to better distinguish the different conditions. As they are in their current form, the results represented are difficult to be interpreted.

Another minor comment regards the figure 1 where one the symbol used doesn’t coincide with the ones described in the figure legend.

Except for these suggestions, I have no other comments to provide.

Author Response

Dear Reviewer,

Thank you for your thoughtful comments that have resulted in a clearer manuscript. Your valuable input was implemented into our revised manuscript, in particular all three figures have been edited according to your suggestions. All changes to the newly submitted manuscript appear in blue font.

Round 2

Reviewer 1 Report

I appreciate the authors´effort to improve the quality of their article, and indeed they did. However, this does not change anything about the fact that this journal is not suitable for this kind of study. The results, without the need to repeat my criticisms about them, are of some use and worth being published. But this should be done in an Orthopedic, sports medicine or radiologic journal. The fact that there will be a special issue on Achilles tendon pathology does not change anything about this basic problem. 

I wish that this work is published, but it should be in a journal that fits this kind of studies better.